

# SMOS near real time soil moisture product: processor overview and first validation results

Nemesio Rodríguez-Fernández[1,2], Joaquin Muñoz Sabater[1], Philippe Richaume[2], Patricia de Rosnay[1], Yann Kerr[2], Clement Albergel[1,3], Matthias Drusch[4], and Susanne Mecklenburg[5]

[1]European Centre for Medium-Range Weather Forecasts, Shinfield Road, Reading RG2 9AX, UK
[2]CESBIO (Université de Toulouse, CNES, CNRS, IRD), 18 av. Edouard Belin, bpi 2801, 31401 Toulouse, France
[3]CNRM UMR 3589, Météo-France/CNRS, Toulouse, France
[4]European Space Agency, ESTEC, Noordwijk, The Netherlands
[5]European Space Agency, ESRIN, Frascati, Italy

*Correspondence to:* nemesio.rodriguez@cesbio.cnes.fr

**Abstract.** Measurements of the surface soil moisture (SM) content are important for a wide range of applications. Among them, operational hydrology and numerical weather prediction, for instance, need soil moisture information in near-real-time (NRT), typically not later than 3 hours after sensing. The European Space Agency (ESA) Soil Moisture and Ocean Salinity (SMOS) satellite is the first mission specifically designed to measure soil moisture from space. The ESA level 2 SM retrieval algorithm
is based on a detailed geophysical modelling and cannot provide SM in NRT. This paper presents the new ESA SMOS NRT SM product. It uses a neural network (NN) to provide SM in NRT. The NN inputs are SMOS brightness temperatures for horizontal and vertical polarizations and incidence angles from 30° to 45°. In addition, the NN uses surface soil temperature from the European Centre for Medium Range Weather Forecasts (ECMWF) Integrated Forecast System (IFS). The NN was trained on SMOS Level 2 SM. The swath of the NRT SM retrieval is somewhat narrower ($\sim$ 915 km) than that of the L2 SM dataset
($\sim$ 1150 km), which implies a slightly lower revisit time. The new SMOS NRT SM product was compared to the SMOS Level 2 SM product. The NRT SM data shows a standard deviation of the difference with respect to the L2 data of $< 0.05$ m$^3$m$^{-3}$ in most of the Globe and a Pearson correlation coefficient higher than 0.7 in large regions of the Globe. The NRT SM dataset does not show a global bias with respect the L2 dataset but can show local biases of up to 0.05 m$^3$m$^{-3}$ in absolute value. The two SMOS SM products were evaluated against in situ measurements of SM from more than 120 sites of the SCAN (Soil Climate
Analysis Network) and the USCRN (United States Climate Reference Network) networks in North America. The NRT dataset obtains similar but slightly better results than the L2 data. In summary, the neural network SMOS NRT SM product exhibits performances similar to those of the Level 2 SM product but it has the advantage of being available in less than 3.5 hours after sensing, complying with NRT requirements. The new product is processed at ECMWF and it is distributed by ESA and via the European Organisation for the Exploitation of Meteorological Satellites (EUMETSAT) multicast service (EUMETCast).





# 1 Introduction

Surface soil moisture (SM) represents less than 0.001 % of the global freshwater budget by volume but it plays an important role in the water, carbon and energy cycles (Lahoz and De Lannoy, 2013). SM is the water reservoir for plants and agriculture and it affects the evolution of diseases such as malaria (Montosi et al., 2012; Peters et al., 2014). The amount of moisture in

the soil is an important variable to understand the coupling of the continental surface and the atmosphere (Koster et al., 2004; Seneviratne et al., 2006; Tuttle and Salvucci, 2016). Surface SM softens the effect of precipitations, affects the partitioning of the water cycle (infiltration and run-off, and therefore the groundwater storage McColl et al. (2017)) and it can can also be used to improve rainfall estimations (Pellarin et al., 2008; Crow et al., 2009; Brocca et al., 2016). SM measurements have been used to perform data assimilation into land surface models (Xu et al., 2015; Blankenship et al., 2016; Lievens et al., 2016), SVAT

(Soil Vegetation Atmosphere Transfer) models (Martens et al., 2016; Ridler et al., 2014; Muñoz Sabater et al., 2007) and in carbon-cycle models (Scholze et al., 2016). SM data assimilation can improve river discharge predictions, and remote sensing measurements are useful in otherwise data-scarce catchments (Alvarez-Garreton et al., 2016; Chen et al., 2011; Pauwels et al., 2002). SM measurements are useful to monitor landslide risks (Hawke and McConchie, 2011) and remotely sensed SM has been used to compute landslide susceptibility maps (Ray et al., 2010).

Regarding flood forecasting, in the framework of the European Flood Awareness System (EFAS) the forecast accuracy improves significantly (5-10%) when remotely sensed SM is assimilated in addition to discharge data (Wanders et al., 2014). SM initial conditions are among the most important hydrological properties affecting flash flood triggering (Norbiato et al., 2008; Ponziani et al., 2012). The assimilation of soil moisture products from the Advanced Scatterometer (ASCAT) has been successfully used in the context of flash flood early warning systems in Mediterranean catchments (Cenci et al., 2016).

In addition to operational hydrology applications, operational numerical weather prediction also benefits from remotely sensed SM data assimilation. Meteorological agencies such as the European Centre for Medium Range Weather Forecasts (ECMWF) and the United Kingdom Met Office assimilate ASCAT surface SM into their operational numerical weather predictions models (de Rosnay et al., 2013; Dharssi et al., 2011). The approach has also been tested in off-line mode at Meteo France (Barbu et al., 2014). To be useful for operational applications, remotely sensed data should be available in near-real-time

(typically less than 3-4 hours after sensing, hereafter NRT).

The Soil Moisture and Ocean Salinity (SMOS) European Space Agency (ESA) satellite (Kerr et al., 2010) is the first instrument that has been specifically designed to measure soil moisture from space. It carries an L-Band (1.4 GHz) radiometer to perform full polarization and multi-angular (0° - 60°) measurements of the Earth thermal emission. ECMWF uses SMOS NRT brightness temperatures in their operational Integrated Forecasting System (Muñoz Sabater et al., 2012). The ESA SMOS

operational Level 2 SM retrieval algorithm is based on a point-per-point iterative minimization of the difference of a physical model and the satellite measurements (Kerr et al., 2012). The free parameters are the soil moisture content and the 1.4 GHz opacity, which is mainly due to the water content of the vegetation in between the soil surface and the sensor (which some authors refer to as VOD, vegetation optical depth).





Many studies have evaluated the SMOS L2 SM dataset in comparison to other remote sensing datasets, models and in situ measurements (Al Bitar et al., 2012; Wanders et al., 2012; Albergel et al., 2012; Bircher et al., 2013; Al-Yaari et al., 2014a, b; Leroux et al., 2014; Louvet et al., 2015; Kerr et al., 2016). SMOS shows very good global performance although other remote sensing and model products can show better performances at some sites. In any case, datasets from the only two instruments specifically conceived to measure SM, SMOS and NASA's Soil Moisture Active Passive (SMAP), compare very well with each other (Jackson et al., 2016; Burgin et al., 2016).

As already mentioned, most operational users over land, in particular in numerical weather prediction and operational hydrology, require SM information to be available in NRT, typically referring to less than 3-4 h after sensing. This requirement cannot be met with the operational SMOS level 2 processor due to the complexity of the geophysical retrieval algorithm and associated processing times. However, with 6 years of SMOS measurements available, statistical algorithms can be exploited to provide faster retrievals and neural networks have been shown to be a promising technique to generate a SM dataset from SMOS brightness temperatures (Rodríguez-Fernández et al., 2015). Based on the latter, a neural network processing chain to provide SMOS SM in NRT has been implemented by ESA. The requirements are that the NRT dataset should display at least the same accuracy as the geophysical level 2 soil moisture data product, the data should be retrieved over a large swath, and the retrieval should rely on a minimum of auxiliary data files. The new NRT product is available from 2016 onwards and it is distributed through the World Meteorological Organization's Global Telecommunication System (GTS) and the European Organisation for the Exploitation of Meteorological Satellites (EUMETSAT) EUMETCast service in NetCDF format. EUMETCast is a dissemination system that uses commercial telecommunication geostationary satellites and research networks to multi-cast data files to a wide user community.

This paper describes the SMOS NRT SM processing chain and discuss the first evaluation results. It is organized as follows. Section 2 describes the data used for the implementation and the validation of the SMOS NRT SM product. Section 3 discusses the NRT SM processing chain (more details are given in the Appendix). Section 4 contains a description of the methods used to evaluate the NRT SM product. Section 5 presents the evaluation results. Finally, a summary is presented in Section 6.

## 2 Data

### 2.1 SMOS satellite

SMOS (Mecklenburg et al., 2012; Kerr et al., 2010) measures the thermal emission from the Earth at a frequency of 1.4 GHz in full-polarization and for incidence angles from $0°$ to $\sim 60°$. The full incidence angle range is accesible in the center of the swath. On the contrary, only angles in the $40°$-$45°$ range are accessible all across the swath. SMOS has 69 antennas to perform interferometry and synthesize an aperture of $\sim 7.5$ m (Anterrieu and Khazaal, 2008). The spatial resolution on the ground, defined as the projection of the full width at half maximum of the synthesized beam, is 43 km on average over the field of view (Kerr et al., 2010). The satellite follows a sun-synchronous polar orbit with 6:00 am/pm equator overpass time for ascending/descending half-orbits.





### 2.1.1 SMOS Level 2 soil moisture

The SMOS Level 2 algorithm is based on the iterative minimization of the difference in between modelled and observed brightness temperatures ($T_b$'s) to retrieve SM and optical depth ($\tau$). The model uses the $\tau - \omega$ (single scattering albedo) approach to account for interaction of L-band radiation with the vegetation (Wigneron et al., 2007). In the case of forest, two

contributions to the opacity are taken into account: one from the arboreous component, which is estimated from the maximum Leaf Area Index (LAI, Ferrazzoli et al., 2002), and another from the understory vegetation. Soil temperature is obtained from ECMWF Integrated Forecast System (IFS) data. For footprints with mixed land cover, the SM content of the minor land cover is estimated from ECMWF IFS and its contribution to the $T_b$ is fixed. For such cases, the SMOS SM retrieval is only performed for the dominant land cover class within the footprint (Kerr et al., 2012). ESA Level 2 SM data are provided in an Icosahedral

Equal Area (ISEA) 4H9 grid (Sahr et al., 2003) with a sampling space of 15 km.

    The version of the SMOS L2 SM dataset used in this study is v620, which became operational in May 2015. In order to have enough data for a robust training of the NN, an additional dataset from 01/June/2010 to 30/June/2012 was reprocessed with the same version v620 of the L2 SM algorithm. The evaluation of the NRT SM product has been done from May 2015 to the time of the NRT SM implementation (end of 2015).

### 2.1.2 SMOS near Real-Time soil moisture

The SMOS Near Real-Time SM product was obtained training a neural network using SMOS $T_b$'s and soil temperature from ECMWF models as input. The training dataset used for the supervised learning phase of the neural network was the SMOS Level 2 SM product. SMOS $T_b$'s are provided by ESA to ECMWF in NRT (less than 3 hours after sensing). The SMOS NRT SM product is computed at ECMWF and delivered to ESA, where the data are sent to EUMETSAT for dissemination

via EUMETCast. The SMOS NRT SM data are provided in NetCDF files in the same ISEA 4H9 grid of other ESA SMOS products. The version of the SMOS NRT SM data used in this study is version 100. More details on the NRT SM processor are presented in Sect. 3 and in the Appendix.

### 2.2 In situ soil moisture measurements

The SMOS NRT SM product was evaluated against in situ measurements of SM over a large number of sites. The same

evaluation was done with the Level 2 SM product. The in situ data used for those evaluations are described below.

    The Soil Climate Analysis Network (SCAN) of the United States Department of Agriculture (Schaefer et al., 2007) has been widely used to evaluate modelled and remote sensing soil moisture datasets and it contains over 100 sensors/sites. The sensors are located in agricultural regions with a relatively homogeneous landscape in many cases. The sensors used in this study are placed horizontally at 5 cm depth.

The United States Climate Reference Network (USCRN, Bell et al., 2013) is a network of climate monitoring stations with sites across the U.S.A., managed and maintained by the National Oceanic and Atmospheric Administration (NOAA). This network was designed with climate science in mind. The stations are placed in pristine environments expected to be free of





**NRT SM** (m³/m³)

**L2 SM** (m³/m³)

**NRT SM error** (m³/m³)

**L2 SMerror** (m³/m³)

**Figure 1.** Comparison of the NRT-NN SM product (a) and the Level 2 SM (b) for one orbit of day 27/May/2012. The corresponding NRT-NN uncertainty is shown in panel (c), while the L2 SM uncertainty is shown in panel (d).

development for many decades. There are around 140 stations with sensors at different depths. The sensors used in this study are horizontally installed at 5 cm.

    The in situ data have been obtained directly from the teams operating both networks but these datasets are also available

from the International Soil Moisture Network Dorigo et al. (2011).





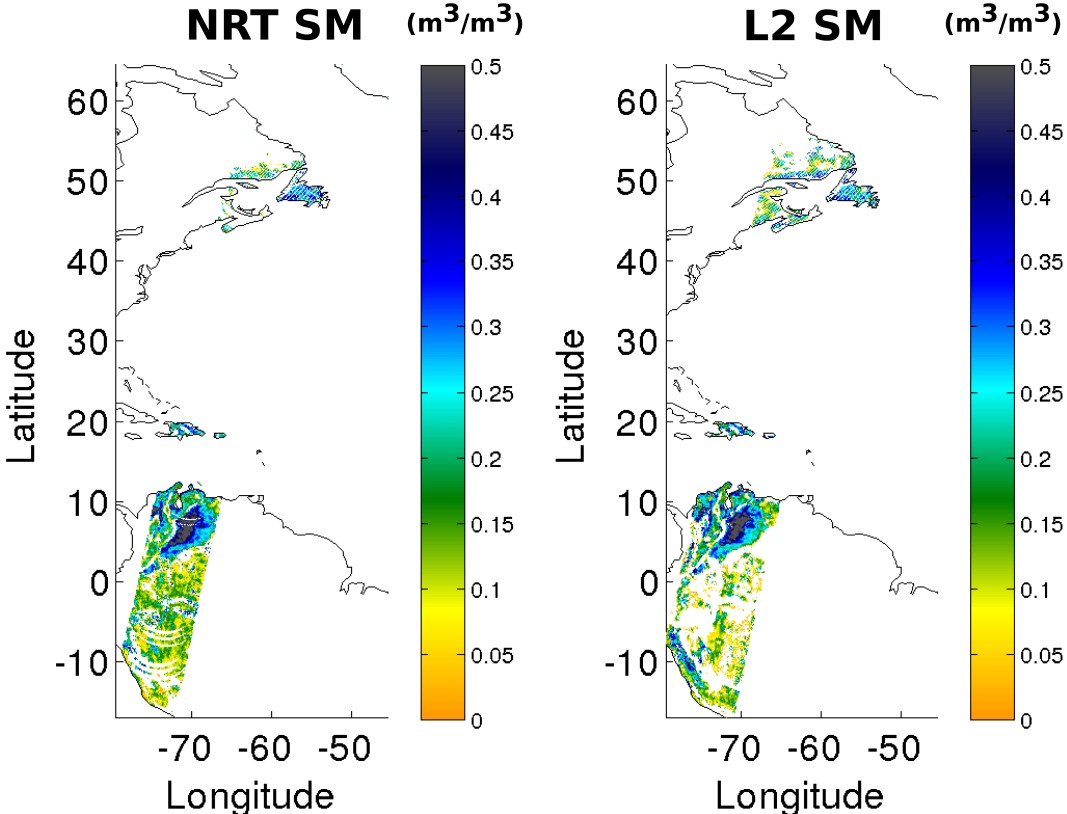

**Figure 2.** Comparison of the NRT-NN SM product (a) and the Level 2 SM (b) for one orbit of day 27/May/2012.

## 3 The SMOS Near-Real-Time soil moisture processor

The SMOS NRT SMOS processor is based on a neural network approach proposed by Rodríguez-Fernández et al. (2015) to retrieve soil moisture from SMOS observations. In that study, SMOS Level 3 $T_b$'s (binned in 5°-width incidence angle bins) were used as input and ECMWF SM modeled fields were used as reference data during the training phase. In the context of the operational NRT SM processor the main input to the neural network are SMOS near-real-time $T_b$'s and the reference dataset used for the training phase is the ESA Level 2 SM dataset. In addition, taking into account operational constrains, the only complementary data used for the retrieval are soil temperature estimations from ECMWF models.

### 3.1 Input data

The input to the SMOS NRT SM processor are SMOS near-real-time $T_b$'s, which are distributed by ESA to ECMWF in BUFR (Binary Universal Form for the Representation of meteorological data) format (Gutierrez and Canales Molina, 2010; de Rosnay et al., 2012). The $T_b$'s are provided with the polarization referred to the antenna reference frame $XY$. Several quality checks are performed to filter the $T_b$'s: $T_{bX}$ and $T_{bY}$ should be in the expected physical range [80 K − 340 K] and the real and



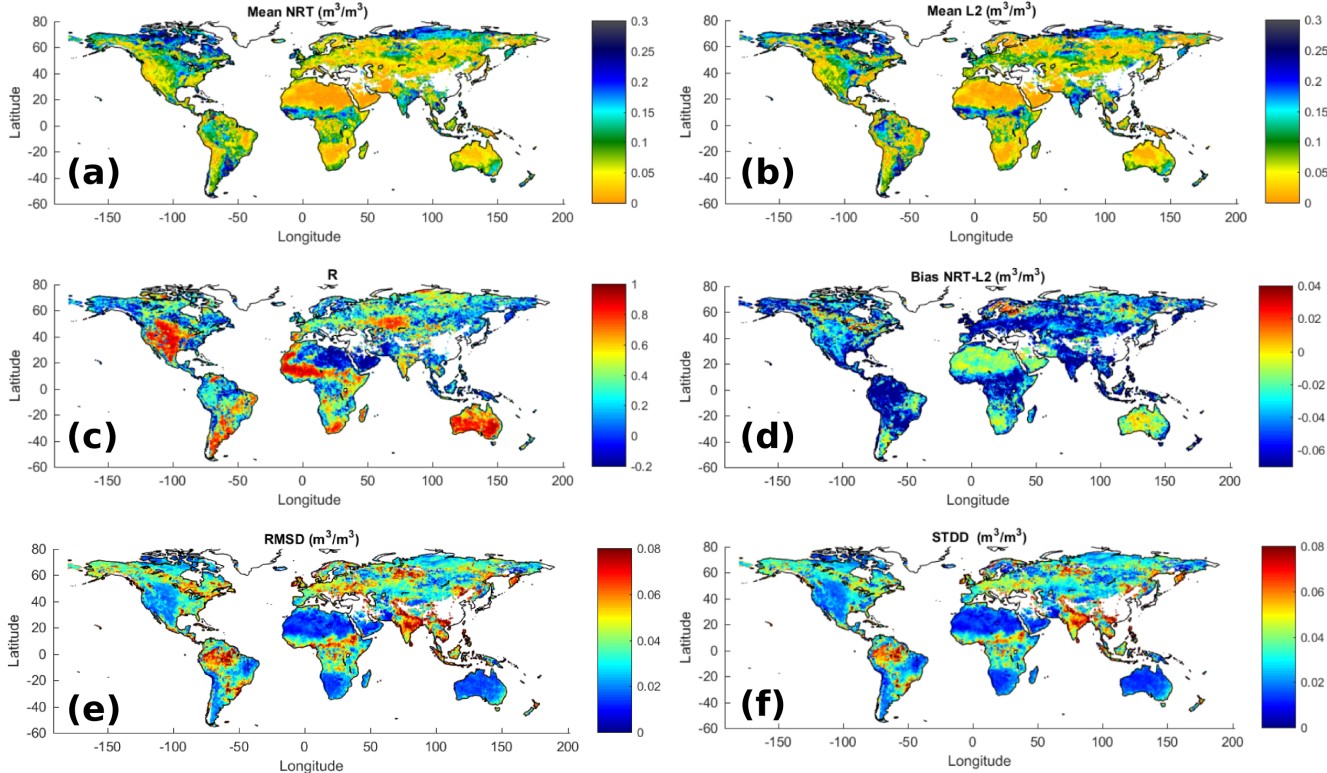

**Figure 3.** Mean SM for the NRT (a) and the L2 (b) SMOS products. Pearson correlation (c), bias (d), root mean square of the difference (e) and standard deviation (f) of the difference of the NRT SM and L2 SM.

imaginary components of the cross-polarised measurements ($T_{\mathrm{b}\,XY}$) should be in the range $[-50\,\mathrm{K}, 50\,\mathrm{K}]$, otherwise the $T_{\mathrm{b}}$'s are considered to be corrupted or affected by RFI (Radio Frequency Interference from human-built equipement). The observed $T_{\mathrm{b}}$'s are filtered out if a specific BUFR flag indicates that the observation is located in a zone affected by the aliased image of the Sun. The selected NRT $T_{\mathrm{b}}$'s are transformed from the antenna-based $XY$ reference frame to the ground-based horizontal and vertical ($HV$) reference frame as described by Al Bitar et al. (2017).

In a second step the $HV$ $T_{\mathrm{b}}$'s are averaged in $5°$-width incidence angle bins. Three angle bins are actually used for training and applying the neural network: $30°-35°$, $35°-40°$, and $40°-45°$. As discussed by Rodríguez-Fernández et al. (2016), using these three angle bins is the best trade off of performances (which improve with a large angular signature) and swath-width of the retrieval (which decreases with an increasing number of angle bins used). With this configuration SM is retrieved in the central 914 km of the swath (the maximum possible swath is $\sim 1150$ km). A SM retrieval can only be done if there is a well-defined value of the $T_{\mathrm{b}}$'s for all the three angle bins and the two polarizations $H$ and $V$. The current implementation of the NRT SM processor does not perform any interpolation of the $T_{\mathrm{b}}$ versus incidence angle profiles.





Using the SMOS $T_b$'s measured at a time $t$ for a given latitude ($\lambda$) and longitude ($\phi$) grid point and for each polarization and incidence angle bin, $T_{b_{\lambda\phi}}(t)$, a local normalized index can be computed as:

$$I_{\lambda\phi}(t) = SM_{\lambda\phi}^{T_b^{min}} + [SM_{\lambda\phi}^{T_b^{max}} - SM_{\lambda\phi}^{T_b^{min}}] \frac{T_{b_{\lambda\phi}}(t) - T_{b_{\lambda\phi}}^{min}}{T_{b_{\lambda\phi}}^{max} - T_{b_{\lambda\phi}}^{min}} \tag{1}$$

Where $T_b^{max}$ and $T_b^{min}$ are the maximum and minimum values of the $T_b$'s in the local time ($\lambda, \phi$) series, $SM^{T_b^{min}}$ and $SM^{T_b^{max}}$ are the associated SM in the SMOS Level 2 SM reference dataset. The index $I$ is computed for each incidence angle bin and polarization at the time $t$ of the SMOS acquisition and it contains a local information on the dynamic ranges of both the measured $T_b$'s and the reference SM. In the current version of the processor (v100), $T_b^{max,min}$ and $SM^{T_b^{max,min}}$ have been computed using data from 01/June/2010 to 30/June/2012 (the same period used to train the neural network, see Sect. 3.2).

The only auxiliary data used by the SMOS SM NRT processor are snow depth and soil temperature from the latest high-resolution forecast produced by the ECMWF IFS, with a typical latency of less than 1 h. The ECMWF IFS soil temperature in the 0–7 cm layer is used as input to the NN, as it increases the performances of the retrieval (Rodríguez-Fernández et al., 2016). SMOS data from a given grid point are not used if snow is found in that point based on the latest ECMWF snow depth forecast field or if the soil temperature forecast of the top soil 7 cm is below 274 K. Finally, a SM retrieval is not provided if more than 50% of the SMOS footprint is covered by water. This filter avoids spurious too high soil moisture values near the coastlines, for instance.

## 3.2 The neural network processor

The HV angle-binned $T_b$'s have been collocated with ECMWF IFS forecasts for the soil temperature and snow cover and finally they have been collocated with version 620 SMOS L2 SM data (Kerr et al., 2012) in the 01/June/2010 to 30/June/2012 period. As discussed above, a local normalized index $I$ has been computed from extreme $T_b$'s and the associated L2 SM. In addition to the filters discussed above, to compute the extreme values tables and for the training of the NN, the following filters have also been applied:

- The latitude is limited to the [-60°,75°] range.

- The fraction of the SMOS footprint occupied by surface water Water is required to be zero.

- A SMOS L2 SM value associated to the maximum or minimum $T_b$ is required (otherwise $I$ cannot be defined).

- The SM uncertainty provided by `Dqx` (Data Quality Index) parameter in L2 SM data files was required to be lower than 0.06 m³m⁻³ to use the most reliable data for the training.

The input vectors contain $T_b$'s and $I$ indexes for H and V polarizations and the three angle bins from 30° to 45° and the soil temperature from 0 to 7 cm from ECMWF IFS forecast. Therefore, input vectors have a total of 13 elements. All the 13 elements must be well-defined to train the NN and there must be a well-defined associated SM value.





One fifth of the vectors in the training data base were selected randomly to have $\sim 3 \times 10^5$ vectors. A subset of 60% of those

vectors is used for the actual training, 20% is used for evaluation of the NN performances during the training and to avoid over-training, the final 20% is used to test the performances of the trained NN *a posteriori*. Gradient back-propagation and minimization with the Levemberg-Marquard algorithm has been used. One single hidden layer with 5 neurons has been used, as it has been shown by Rodríguez-Fernández et al. (2016) that it is enough to capture the relationship between the input data and the reference SM and while keeping the NN as simple as possible. No signs of overtraining were found and the training

was stopped after 50 iterations when the mean squared difference was asymptotically approaching to a minimum. When the trained NN was applied to the test subset and the NN output was compared to the SMOS L2 SM, the Pearson correlation R was 0.86, the standard deviation of the difference (STDD) was 0.068 $m^3/m^3$ and the Root Mean Square Error or Difference (RMSE) was also 0.068 $m^3/m^3$, which implies that there was not a significant bias in between both SM datasets. This results show that the NN ability to capture the dynamics of the current L2 SM dataset is very good. The evaluation results discussed

in Sect. 5 below confirm that the quality of the SM-NRT-NN product fulfil the specifications of the operational product.

NN NRT SM uncertainties were computed by error propagation through the neural network taking into account the error of the $T_\mathrm{b}$'s used as input as explained in the Appendix.

### 3.3 SMOS NRT SM processor output

The SMOS NRT SM product is a land-only product, collocated and delivered in the ISEA 4H9 grid (Sahr et al., 2003) common

to other ESA SMOS products. The main characteristics of the product and the description of the fields are presented in Muñoz-Sabater et al. (2016). The processor output fields are:

- The ISEA grid point number

- Latitude

- Longitude

- Year

- Month

5   - Day

- Seconds from midnight (all times are UT)

- NRT soil moisture

- Soil moisture uncertainty

- RFI probability

10   Figure 1 shows the NRT-NN SM product and its associated uncertainty for a portion of an orbit of day 27/May/2012.





**Table 1.** Comparison to in situ measurements over the USCRN and SCAN networks. The columns are: the SM product, the mean number of points in the time series, the mean and median Pearson correlation with respect to in situ measurements, the mean bias (mean in situ SM minus mean SMOS SM), the RMS and STD of the difference time series averaged over all sites, and the Pearson correlation of the anomalies time series ($R_a$). The statistics have been computed independently for the SM-NRT-NN and the SM-L2 product. The number of SM retrievals is, on average, larger for the SM-L2. The two lower rows show the results using only times for which both the SM-NRT-NN and the SM-L2 products are simultaneously available.

| SM | Mean $N_{pts}$ | Mean R | Median R | mean Bias | mean RMSD | Mean STDD | mean $R_a$ |
|------|------|------|------|------|------|------|------|
| L2 | 186 | 0.63 | 0.64 | 0.035 | 0.100 | 0.065 | 0.56 |
| NRT | 94 | 0.70 | 0.71 | 0.036 | 0.095 | 0.058 | 0.48 |
| L2 | 88. | 0.67 | 0.69 | 0.026 | 0.092 | 0.062 | 0.59 |
| NRT | 88. | 0.71 | 0.72 | 0.031 | 0.091 | 0.056 | 0.56 |

# 4 Methods

## 4.1 Global evaluation

Several metrics have been used to evaluate the NRT SM dataset from 15/May/2015 to 25/November/2015 against the SMOS L2 SM dataset. For all grid points $\lambda\phi$, the temporal means of both SM dataset, $\overline{SM}_{\lambda\phi}^{L2}$ and $\overline{SM}_{\lambda\phi}^{NRT}$, have been computed as:

$$\overline{SM}_{\lambda\phi}^{NRT} = \frac{1}{N_t} \sum_{i=1}^{N_t} SM_{\lambda\phi}^{NRT}(t_i) \qquad (2)$$

and

$$\overline{SM}_{\lambda\phi}^{L2} = \frac{1}{N_t} \sum_{i=1}^{N_t} SM_{\lambda\phi}^{L2}(t_i), \qquad (3)$$

using only times ($t_i$) for which a well-defined value is present simultaneously in both datasets. This number is in principle different for each $\lambda\phi$ grid point, but it will be noted as $N_t$ in the following instead of $N_{\lambda\phi_t}$ to simplify the notation.

A bias map has been computed from the local ($\lambda$ and $\phi$) mean of each dataset as follows:

$$Bias_{\lambda\phi} = \overline{SM}_{\lambda\phi}^{NRT} - \overline{SM}_{\lambda\phi}^{L2}. \qquad (4)$$





In order to compare the temporal dynamics of the two datasets, the Pearson correlation $R$ has also been computed as follows:

$$R_{\lambda\phi} = \frac{\sum\limits_{i=1}^{N_t}(SM_{\lambda\phi}^{NRT}(t_i) - \overline{SM}_{\lambda\phi}^{NRT})(SM_{\lambda\phi}^{L2}(t_i) - \overline{SM}_{\lambda\phi}^{L2})}{\sqrt{\sum\limits_{i=1}^{N_t}(SM_{\lambda\phi}^{NRT}(t_i) - \overline{SM}_{\lambda\phi}^{NRT})^2}\sqrt{\sum\limits_{i=1}^{N_t}(SM_{\lambda\phi}^{L2}(t_i) - \overline{SM}_{\lambda\phi}^{L2})^2}}, \tag{5}$$

where the sum runs for all the points available at a given position $\lambda\phi$: $N_t$.

The absolute values of the two datasets have been evaluated using the standard deviation of the difference as a metric. The local time series difference $D$ of the two datasets was defined as:

$$D_{\lambda\phi}(t) = SM_{\lambda\phi}^{NRT}(t) - SM_{\lambda\phi}^{L2}(t) \tag{6}$$

The standard deviation of the difference time series (STDD) has been computed as:

$$STDD_{\lambda\phi} = \sqrt{\overline{D_{\lambda\phi}^2} - \overline{D_{\lambda\phi}}^2} = \sqrt{\frac{1}{N_t}\sum_{i=1}^{N_t}D_{\lambda\phi}^2(t_i) - \left(\frac{1}{N_t}\sum_{i=1}^{N_t}D_{\lambda\phi}(t_i)\right)^2} \tag{7}$$

In some studies, the STDD is calculated indirectly from the bias and the root mean squared difference (RMSD) and called unbiased-RMSD (ubRMSD).

## 4.2 Local evaluation against in situ measurements

The SMOS NRT SM and the SMOS L2 SM datasets have been evaluated against the in situ measurements discussed in Sect. 2 in a consistent manner. First, for each station available, a quality check of the data was performed. Sites with suspicious data
(e.g. measurements discontinuity, spurious jumps) were eliminated. The locations of the 127 retained stations is showed in Fig. 4.

    The same metrics discussed in the previous section have been computed for the NRT SM dataset with respect to the in situ measurements and for the L2 SM dataset with respect to the in situ measurements. The Pearson correlation was used to compare temporal dynamics of two SM datasets. The long term (seasonal) dynamics were compared by computing the Pearson
5  correlation coefficient R of $SM^{L2}$ and $SM^{NRT}$ with respect to $SM^{inSitu}$, site per site. In addition, the short-scale dynamics were evaluated by computing site per site the Pearson correlation of the anomalies times series. Following Albergel et al. (2009), the SM anomaly at given time $t$, $SM_a(t)$, was computed using a 31 day window centred at $t$ as follows:

$$SM_a(t) = \frac{SM(t) - \text{Mean}(SM(t-15, t+15))}{\text{STD}(SM(t-15, t+15))} \tag{8}$$

where $SM(t-15, t+15)$ represents the ensemble of measurements in the 31-day window. The Pearson correlation coefficient
10  $R$ computed with the anomalies time series will be referred to as $R_a$ in the following.





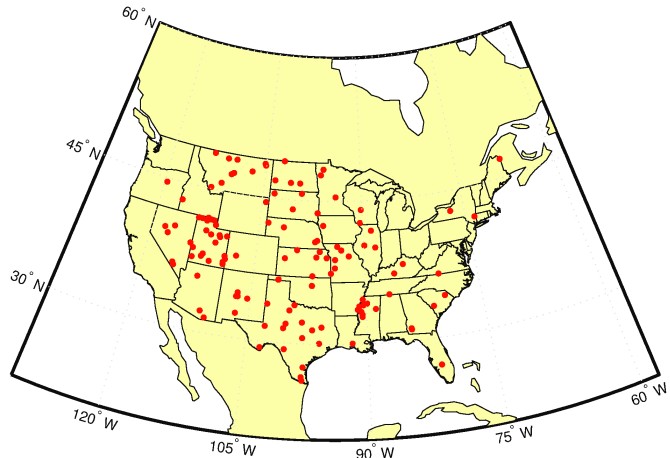

**Figure 4.** Location of the in situ measurements sites used in this study.

The metrics were computed independently for the NRT and the L2 datasets in a first step. In a second step, the metrics were recomputed only using times for which both the NRT and the L2 were simultaneously available, and thus, using the same number of points for the two time series.

## 5   Results: SMOS NRT soil moisture evaluation

### 5.1   Swath-level comparison to SMOS L2 SM

Figures 1a,c show the NRT-NN SM product and its associated uncertainty for of a portion of an orbit of day 27/May/2012. The corresponding L2 SM and its associated uncertainty as given by the `DQX` (Data Quality Index) parameter (Kerr et al., 2012) are also shown (Figs. 1b,d). As discussed in Sect. 3, the swath width of the NRT-SM retrieval is somewhat narrower than the L2 SM one but both maps show similar spatial structures and numerical values. The uncertainties have similar numerical values as well, but the spatial patters are not the same. This is expected as the two retrieval algorithms are different. Finally, it should be noted that the spacial coverage can be different for both products as shown in Fig. 2:

- the NRT SM product can show circle-arc gaps when not all of the angle bins have a well defined $T_{\mathrm{b}}$ value, while in contrast the L2 algorithm can perform an inversion even if some $T_{\mathrm{b}}$'s have been filtered out.

- The NRT SM global retrieval algorithm can provide a SM estimate even when the local minimization of the L2 algorithm does not converge. This can happen mainly in dense forest areas.





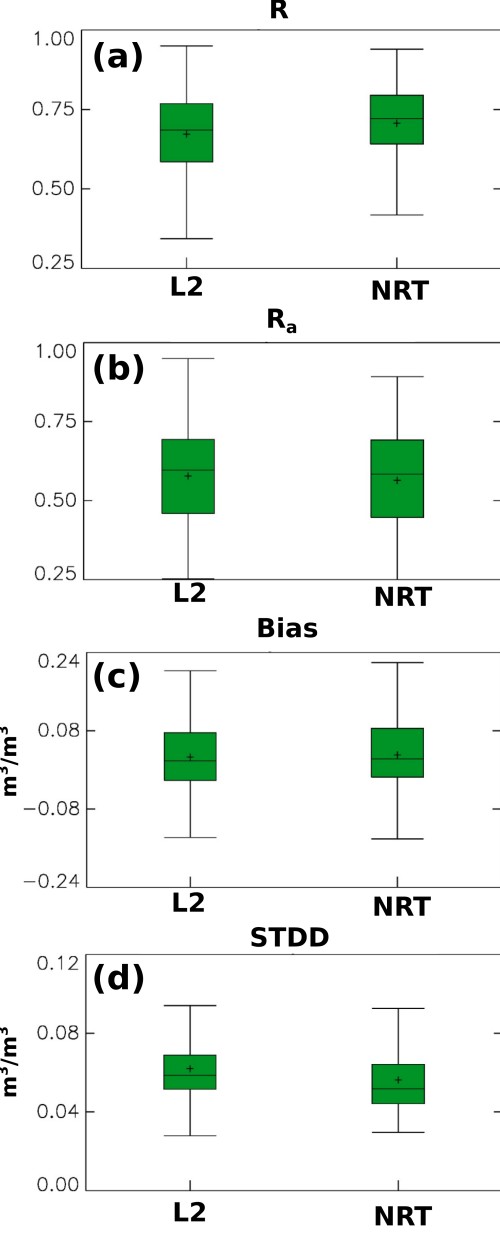

**Figure 5.** Boxplots for (a) the Pearson correlation coefficient ($R$) of the NRT and L2 time series with respect to the in situ measurements (b) Pearson correlation coefficient of the anomalies time series ($R_a$), (c) Bias (mean *in situ* minus mean SMOS SM), and (d) STDD of the two SMOS products in comparison to in situ measurements. The box contains the middle 50% of the data, the central bar represents the median value of the distribution. The upper edge (hinge) of the box indicates the 75th percentile of the data set ($q_3$), and the lower hinge indicates the 25th percentile ($q_1$). The mean values are also shown as black crosses. The upper and lower bars represent the minimum and maximum values of the distribution excluding outliers. Points are considered as outliers if they are larger than $q_3 + 1.5(q_3 - q_1)$ or smaller than $q_1 - 1.5(q_3 - q_1)$.





## 5.2 Global evaluation with respect to SMOS L2 SM

The SMOS NRT SM product has also been compared to the SMOS L2 SM product globally and over the period mentioned in Sect. 4. Figs. 3a,b show the mean of the NRT and L2 SM products over the period of the study. Both maps show an overall excellent agreement, although it is possible to appreciate a significant negative bias ($-0.05$ m$^3$m$^{-3}$) in the NRT SM product in the regions with the highest L2 SM (tropical and boreal forest) . The typical number of points with both NRT-SM-NN and SM-L2 in the evaluation period is $\sim 100$. The correlation of both products is high ($> 0.7$) over a large part of North-America, the southernmost part of South-America, the Iberian peninsula, the Sahel and South-Africa, Australia and parts of central Eurasia. The correlation is significantly lower over forest (both tropical and boreal) and in deserts such as the Sahara. In the Sahara the low correlation is probably not significant because the SM values are very low and the variance is driven by the noise. Actually, Fig. 3f shows that the STDD is also very low in this region. Therefore, L2 SM and NRT SM have actually similar values. In contrast, dense forest regions show a high STDD in addition to a low R. Therefore, both products show some differences in these regions. Unfortunately, in situ measurements are not available to perform an independent evaluation of both dataset for dense forest sites. In conclusion, both products show similar dynamics over large parts of the Globe. The Bias map (Fig. 3d) shows that the NRT SM products shows a tendency to underestimate the L2 SM dataset, which is a expected behaviour as it has been obtained using a regression technique and extreme values are under-represented in the reference dataset. The most significant effect of the bias is to increase the RMSD (Fig. 3e) with respect to the STDD in parts of Europe and Canada. However, one should note that both the RMSD and the STDD are lower than 0.04 m$^3$m$^{-3}$ over most of the Globe (all except the reddish regions in Fig. 3e,f).

## 5.3 Evaluation with respect to in situ measurements

The SMOS NRT SM product was evaluated against in situ measurements from the SCAN Schaefer et al. (2007) and USCRN Bell et al. (2013) networks (Sect. 2). These networks of in situ measurements have been extensively used for the validation of remote sensing data (Albergel et al., 2009; Rodríguez-Fernández et al., 2015; Al Bitar et al., 2012; Albergel et al., 2012; Kerr et al., 2016).

The quality metrics discussed in Sect. 4 have been computed site per site independently for the SMOS NRT. The same evaluation was done for L2 products. The mean number of points in the time series from May 2015 to November 2015 is 186 for the L2 product while is only half of that value for the NRT product. The reason is a longer revisit time of the SM-NRT-NN product due to the narrower swaths of the retrievals and the lack of retrievals if not all the 6 $T_b$'s are well defined for both polarizations and the three angle bins from 30° to 45°.

Table 1 summarizes the results in the form of averages over all the sites (for the Pearson correlation also the median value is given). Both SMOS products show a similar mean bias with respect to the in situ measurements, while the mean STDD and RMSD are slightly lower for the NRT SM product. In order to get further insight into the intrinsic quality differences of both datasets, the same statistics have been computed but only using times for which both SMOS products are retrieved. The results



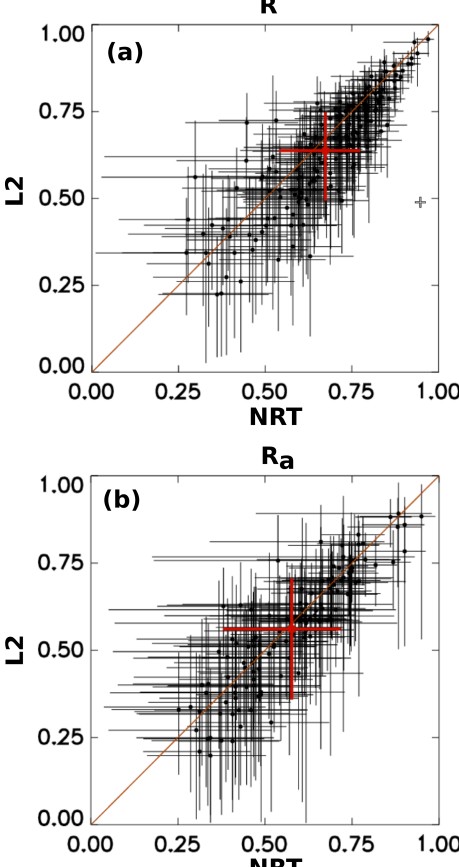

**Figure 6.** (a) Scatter plots showing the Pearson correlation coefficient of the NRT and L2 SM time series with respect to the in situ measurements (R). The errorbars account for the 95% confidence intervals. The red symbols represents an averaged value. (b) Same as (a) but for the anomalies time series ($R_a$).

are also shown in Table 1. The differences in the evaluation of both products decreases, but the NRT product still shows a larger correlation and lower STDD with respect to the in situ measurements than the L2 product.

Since the mean or median values alone do not show the full picture of the evaluation for more than 100 sites, Figs. 5a,b show boxplots for the Pearson correlation coefficient of the time series ($R$) and the anomalies times series ($R_a$), respectively. Figures 5c,d show boxplots for the bias and the STDD. As expected, there is a large variation from one site to another. The bias and STDD distributions are similar for both products. The correlation is as high as almost 1 for some sites both for the NRT SM and L2 SM (the maximum is slightly higher for the later). Interestingly, the lower values of the distribution of the correlation are higher for the NRT product.





Finally, Fig. 6 shows scatter plots of the correlation for the time series and for the anomalies time series taking into account the respective confidence intervals. For most of the sites, both products show the same statistics with respect to the in situ measurements and globally, the scatter plot points lie close to the 1:1 line.

## 6  Conclusions

This paper describes the the SMOS NRT SM processor and the first evaluation of this new operational dataset. This processor
is based on a neural network algorithm similar to that described by Rodríguez-Fernández et al. (2015). The neural network uses SMOS NRT brightness temperatures and ECMWF IFS soil temperature in the 0–7 cm layer as input. It has been trained with SMOS Level 2 SM data as reference. The SMOS NRT brightness temperatures have been transformed from the antenna reference frame to the ground reference frame to express the polarization as horizontal and vertical components. In addition they have been binned in 5°-width incidence angle bins. Soil temperature and snow cover forecasts from ECMWF IFS are used
to filter out frozen soil or soil covered by snow. The uncertainties of the NRT SM data were estimated from the input brightness temperature uncertainties.

The SMOS NRT SM product was evaluated with respect to the original SMOS Level 2 SM product using several months of data. The NRT SM product compares well with the L2 product. The most significant difference is that the NRT SM dataset shows local negative bias at the positions were the highest SM values were found (basically under tropical forest).

The SMOS NRT SM product was also evaluated with respect to in situ measurements of SM over the SCAN and USCRN networks. The NRT product shows similar performances to those of the L2 product. Actually, the mean and median correlation are slightly higher than those obtained for the L2 product. In addition, the standard deviation of the difference with respect to the in situ measurements is lower for the NRT product than for the L2 product.

In summary, the SMOS NRT SM product shows similar performance to the Level 2 product but it has the advantage to be
available in NRT. NRT brightness temperatures are received by ECMWF from ESA in less than three hours after sensing. The NRT SM production takes on average 15 minutes (the arrival of new NRT TBs is checked every 30 minutes and the actual NRT SM production takes a few minutes). The SMOS NRT SM product is delivered to ESA and EUMETSAT for dissemination via EUMETCast. Therefore, the SMOS NRT SM data are available for a large range of operational applications such as numerical weather prediction, hydrological forecast and crop modelling.

5  *Data availability.* The datasets used in this study (Sect. 2) are publicly available. The SMOS L2 SM and NRT SM data can be downloaded from ESA. The SMOS NRT SM data is also available via EUMETCast in NRT. The in situ measurements can be downloaded from the International Soil Moisture Network.





## Appendix A:  NRT SM algorithm

The SMOS NRT has been described qualitatively in Sect. 3. The current section describes the algorithm and the output un-
certainties calculation in detail. Complementary information can be found in Rodríguez-Fernández et al. (2016) and Muñoz-
Sabater et al. (2016).

### A1   Neural network specification

The NN discussed in Sect. 3 has two layers. The first layer contains $j = 1, ..., n_{L1}$ nodes or neurons with an hyperbolic tangent
as activation function. The second layer contains a single neuron with a linear function as activation function. The number of
input elements $n_{in}$ is 13: 6 $T_{\mathrm{b}}$'s (H and V for incidence angle bins from 30 to 45°), 6 index $I$ (H and V for incidence angle
bins from 30 to 45°), and ECMWF soil Temperature. The inputs range should be re-normalized to have values in the $[-1, 1]$
range. If for each input vector element, the minimum and maximum values found during the training phase are given by the
vectors $v_i^{min}$ and $v_i^{max}$ $(i = 1, ..., n_{in})$, the normalization can be computed as follows:

$$v_i^{norm} = -1 + 2 \frac{v_i - v_i^{min}}{v_i^{max} - v_i^{min}}, \ \forall i = 1...n_{in} \tag{A1}$$

The normalized input, together with the first layer weights $(W_{L1})$ and bias $B_{L1}$ are used to compute the first layer outputs
$v^{L1}$ as follows:

$$v_j^{L1} = \tanh(\sum_{i=1}^{n_{in}} W_{L1}^{ij} v_i^{norm} + B_{L1}^{j}), \ \forall j = 1...n_{L1} \tag{A2}$$

The output of the second layer is computed from the first layer outputs, and the second layer weights $(W_{L2})$ and bias $B_{L2}$
as follows:

$$v^{L2} = \sum_{j=1}^{n_{L1}} W_{L2}^{j} v_j^{L1} + B_{L2} \tag{A3}$$

The values of the weights $W_{L1}$ and $W_{L2}$ and the bias $B_{L1}$ and $B_{L2}$ are determined after the training phase. The exact values
for the operational NRT SM processor can be found in Muñoz-Sabater et al. (2016). Finally, to obtain the NN output $(v^{out})$,
the output of the second layer has to be re-normalized as follows:

$$v^{out} = v_{newMin}^{L2} + \frac{v_{newMax}^{L2} - v_{newMin}^{L2}}{v_{oldMax}^{L2} - v_{oldMin}^{L2}} (v^{L2} - v_{oldMin}^{L2}); \tag{A4}$$





## A2 Neural network output uncertainties

From the definition of $I_{\lambda\phi}(t)$ (Eq. 1) it is possible to compute the uncertainties from the $T_b$'s and SM uncertainties. First, Eq. 1 can be rewritten as:

$$I_{\lambda\phi}(t) = SM_{\lambda\phi}^{T_b^{min}} + [SM_{\lambda\phi}^{T_b^{max}} - SM_{\lambda\phi}^{T_b^{min}}] \times I_{1_{\lambda\phi}}(t) \tag{A5}$$

where $I_{1_{\lambda\phi}}(t)$ is given by:

$$I_{1_{\lambda\phi}}(t) = \frac{T_{b_{\lambda\phi}}(t) - T_{b_{\lambda\phi}}^{min}}{T_{b_{\lambda\phi}}^{max} - T_{b_{\lambda\phi}}^{min}} \tag{A6}$$

The uncertainties $\Delta I_{\lambda\phi}(t)$ and $\Delta I_{1_{\lambda\phi}}(t)$ can be computed from uncertainties in $T_b$'s, in the maximum and minimum $T_b$'s and the associated SM values as follows:

$$\Delta I_{\lambda\phi}^2(t) = [SM_{\lambda\phi}^{T_b^{max}} - SM_{\lambda\phi}^{T_b^{min}}]^2 \big(\Delta I_{1_{\lambda\phi}}(t)\big)^2 + \tag{A7}$$

$$+ [1 - I_{1_{\lambda\phi}}(t)]^2 \big(\Delta SM_{\lambda\phi}^{T_b^{min}}\big)^2 + \tag{A8}$$

$$+ [I_{1_{\lambda\phi}}(t)]^2 \big(\Delta SM_{\lambda\phi}^{T_b^{max}}\big)^2 \tag{A9}$$

Where $\Delta I_{1_{\lambda\phi}}(t)$ is given by:

$$\Delta I_{1_{\lambda\phi}}^2(t) = \frac{\Delta T_{b_{\lambda\phi}}(t)^2}{T_{D_{\lambda\phi}}^2} + \tag{A10}$$

$$+ \frac{(\Delta T_{b_{\lambda\phi}}^{max})^2}{T_{D_{\lambda\phi}}^2} \left(\frac{T_{m_{\lambda\phi}}(t)}{T_{D_{\lambda\phi}}}\right)^2 + \tag{A11}$$

$$+ \frac{(\Delta T_{b_{\lambda\phi}}^{min})^2}{T_{D_{\lambda\phi}}^2} \left(-1 + \frac{T_{m_{\lambda\phi}}(t)}{T_{D_{\lambda\phi}}}\right)^2 \tag{A12}$$

as a function of the uncertainty of the local instantaneous measurement $\Delta T_{b_{\lambda\phi}}(t)$ and the uncertainties of the local extreme $T_b$'s values ($\Delta T_{b_{\lambda\phi}}^{max}$ and $\Delta T_{b_{\lambda\phi}}^{min}$).

The uncertainties of the NN output given by Eqs. A1-A4 can be estimated from the uncertainties in the input vector elements ($\Delta v_i$) as follows. First the uncertainties of the normalized input vector can be computed as:

$$\Delta v_i^{norm} = 2\frac{\Delta v_i}{v_i^{max} - v_i^{min}}, \forall i = 1...n_{in} \tag{A13}$$

Using those quantities, the uncertainty of the two layers neural network given by Eqs. A2 and A3 can be expressed as:

$$(\Delta v^{L2})^2 = \sum_{i=1}^{n_{in}} \left\{ (\Delta v_i^{norm})^2 \left(\sum_{j=1}^{n_{L1}} W_{L2}^j W_{L1}^{ij} \sigma^j\right)^2 \right\} \tag{A14}$$





where $\sigma^j$ is given by:

$$\sigma^j = 1 - \tanh^2(\sum_{i=1}^{n_{in}} W_{L1}^{ij} v_i^{norm} + B_{L1}^j), \ \forall j = 1...n_{L1} \tag{A15}$$

10    It is worth noting that in the current implementation the neural network weights are assumed to be constant after training. There exist some methods to estimate the additional output uncertainty that originates from the neural network weight uncertainty that comes from the uncertainties in the reference data used for the training (see for instance Aires et al., 2004) but they are too complex to be implemented in the SMOS NRT SM operational processor. In contrast, some uncertainties in the reference data used for the training have already been taken into account trough $\Delta SM_{\lambda\phi}^{T_b^{min}}$ and $\Delta SM_{\lambda\phi}^{T_b^{max}}$ in Eq. A9.

Finally, the uncertainty after the normalization of the output can be written as:

$$\Delta v^{out} = \frac{v_{newMax}^{L2} - v_{newMin}^{L2}}{v_{oldMax}^{L2} - v_{oldMin}^{L2}} \Delta v^{L2}; \tag{A16}$$

Expressing the output uncertainty as Eq. A14 implies that the vector elements $v_i$ are independent. However, when using index $I$ as input as well as the actual $T_b$'s, some elements are not independent. Since the uncertainties in Eq. A14 are expressed in quadratic form, Eq. A14 gives an upper limit to the output uncertainty.

*Author contributions.* NJRF and JMS are the principal authors of this manuscript. The neural network approach and the uncertainties calculation were designed by NJRF and PR. JMS, PR and NJRF implemented the operational version of the SMOS NRT SM algorithm. The global

evaluation and the comparison to in situ measurements have been done by NJRF and CA, respectively. YK, PdR, MD and SM reviewed the system design and the results.

*Competing interests.* The authors declare that they have no conflict of interest.

*Acknowledgements.* The authors acknowledge useful discussions with Filipe Aires and Catherine Prigent on neural network algorithms to retrieve soil moisture from microwave observations.





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
