# Peer review of "SMOS near real time soil moisture product: processor overview and first validation results"

_Hydrology and Earth System Sciences, 2017_

## Referee Comment (RC1) · Nemesio Rodríguez-Fernández et al. · 26 May 2017

The manuscript reports a new effort in developing an SMOS NRT SM product by using a neural network approach. The NN algorithm uses six SMOS brightness temperatures at incidence angles from 30 to 45 deg binned at 5 deg intervals for horizontal and vertical polarizations and ECMWF IFS soil temperature at 0-7 cm depth. Snow depth and soil temperature at 0-7 cm depth < 274K from ECMWF IFS are also used to exclude certain (snow covered and frozen) situations.

The NRT SM product was compared to ESA SMOS L2 SM product and to the in-situ data from the SCAN and the USCRN soil moisture networks in USA and satisfactory statistics were reported. The used ECMWF soil temperature is from the layer of 0-7 cm, and the in-situ soil moisture sensors are installed at 5 cm depth.

[Figure]

Major comments

The used methods, data and assumptions are described in sufficient details with results of comparison reported and conclusions drawn. However in discussion of the results, some more in depth analysis of the differences and uncertainties between the different products would be very useful for the use of the current product.

1. The depth of the retrieved SM SMOS NRT SM used ECMWF SM at 0-7 cm to train the NN, thus its retrieved SM should represent SM at the same depth, while ESA SMOS L2 SM represents that of the emission depth or sensing depth. In discussing the differences in between both products and those to the in-situ SM which is measured by sensors installed at 5 cm depth and represents an averaged SM around 5 cm, it would be important to point out such issues. In particular it would be important to explain why the correspondence is less good where very low correlation coefficients are reported and where and when such cases occur.

2. The effective soil temperature SMOS NRT SM uses ECMWF 0-7 cm soil moisture as its effective soil temperature. Previous studies have concluded that the model temperature at this depth is not the most adequate one to use (e.g. Dente et al., 2014) and this would increase the uncertainties in particularly semi-arid and arid areas (e.g. Lv et al., 2016). It is also noticed that the SMOS L2 SM uses the Wigneron soil effective temperature. The authors need provide an analysis to settle this issue.

References:

Dente, L., et al. "Combined use of active and passive microwave satellite data to constrain a discrete scattering model." Remote Sensing of Environment 155 (2014): 222-238.

Lv, Shaoning, et al. "A reappraisal of global soil effective temperature schemes." Remote Sensing of Environment 183 (2016): 144-153.

Technical comments

While the manuscript is written in clear language, many typos need to be corrected. Some are listed as follows:

P6L11: constrains -> constraints

P9L12: Levemberg-Marquard -> Levenberg-Marquardt

P9L18: this results -> these results

P11L5: please explain what is 'short-scale dynamics'

P12L16: for of -> for

P12L5: patters -> patterns

P12L6: spacial -> spatial

P14L24: shows –> show

P19L14: taken into account trough -> taken into account through

---

## Referee Comment (RC2) · Anonymous Referee #2 · 15 Jun 2017

This article details the performance of a neural network approach to soil moisture retrieval from SMOS brightness temperature measurements and ECMWF temperature estimates. Comparisons are made to the operational level 2 product and field measurements in northern America. A similar performance to the operational product is demonstrated, with a considerably lower lead time.

The training and validation data appear to have been drawn from the same time period - June 2010-2012. This prompts questions as to the applicability of the approach to data acquired at different times – if the system is only trained and validated for a certain temporal range, how does it perform on data outside that range? Five years

of observations have been acquired since then, validation using some of these would address whether changes on the Earth's surface such as vegetation growth have an impact on accuracy, and whether the neural network approach is reliable when given inputs outside its trained range.

Grammatical and spelling issues are detailed below. Figure 6(a) has either an outlier at the centre-right of the figure which needs to be explained, or a cursor which should be removed. Spelling - accesible should be accessible, "equipement" should be "equipment" Usage "arboreous" is not the right word; possibly "arboreal" was intended. p.8, l. 32 repetition of "water" p. 9, l. 18 "this results" should be "these results" p.12 l.7 "well defined" should be "well-defined" p.16 l.14 remove additional "the"

---

## Referee Comment (RC3) · W. Wagner (Referee) · 26 Jun 2017

GENERAL COMMENTS

This is a well written and interesting paper describing a neural network approach for retrieving soil moisture from SMOS in near-real-time. The results are highly relevant for operational applications in hydrology, meteorology and other earth sciences. The results are realistic and I recommend publication after minor revisions.

SPECIFIC COMMENTS

Page 3, lines 8-9: Explain why the NRT requirements cannot be met by the operational

SMOS Level 2 processor. Is it just a matter of timeliness?

Page 3: Please also discuss possible disadvantages of the neural network approach already here. One topic is certainly the difficulties caused by changes in the sensor characteristics and Level 1 algorithms. Refer to experiences from other operational NRT services.

Page 4, line 5: What exactly do you mean by "arboreous component"? Do you mean the forest canopy?

Page 4, line 16: "... was obtained by training ..."

Page 8, line 23: 50 % is a very large value. Please explain.

Chapters 3.1 and 3.2: Please explain why the neural network relies on normalised data instead of the absolute brightness temperature values.

Page 8, line 32: Here you allow no open water (0 %), which is in stark contrast to the 50 % threshold from above.

Page 8, bottom: Reformat list of put into table.

Chapter 4.1: Please explain why you decided not to use more advanced metrics.

Discussion of Table 1 and Figure 5: The fact that the correlation R is overall slightly better for NRT than the L2 processor is noteworthy. Please discuss this in more detail and provide possible hypothesis why this is the case.

Page 16, line 15: Explain what you mean by "similar". What are the differences in implementation to the approach introduced by Rodriguez-Fernandez (2015)?
* * *

---

## Author Comment (AC1) · 27 Jul 2017

**Referee 3**

GENERAL COMMENTS
This is a well written and interesting paper describing a neural network approach for retrieving soil moisture from SMOS in near-real-time. The results are highly relevant for operational applications in hydrology, meteorology and other earth sciences. The results are realistic and I recommend publication after minor revisions.

We thank you the referee, Prof. Wagner, for his constructive comments that have allowed us to improve the manuscript, in particular putting the new SMOS NRT SM in the broader context of other NRT SM processing chains such as the ASCAT one. We answer below the specific comments.

SPECIFIC COMMENTS
Page 3, lines 8-9: Explain why the NRT requirements cannot be met by the operational SMOS Level 2 processor. Is it just a matter of timeliness?

The operational SMOS level 2 processor performs a very detailed modelling of the Earth emission at 1.4 GHz at two polarizations and a large number of incidence angles. The surface is modelled at 4 km resolution and the processor includes the aggregation of the contributions of those 4 x 4 km^2 cells within a 123 x 123 km^2 square. The processor included simulations of the instrumental response to model the signals that are detected by SMOS. The vegetation optical depth and the soil moisture content are considered as free parameters that vary to minimize the difference of the simulated Tbs and those that were actually been measured by SMOS. This detailed modelling and minimization is done for every position of the ISEA grid with a 15 km spacing, i.e., for a half-orbit those operations have to be repeated ~1e5 times. Currently, the inversion of a half orbit takes around 6 hours using a cluster of computers, which is too long to be useful for NRT applications. In contrast, in the case of the NN algorithm, the training is done once, offline, performing a minimization that uses at once ~5e5 samples. But once that the NN is trained, the inversion of a whole orbit takes a few minutes using a single CPU.
A sentence has been rephrased in the introduction and the reader is referred to Sect. 2, where more detailed information on the level 2 processor is given in the new manuscript.

Page 3: Please also discuss possible disadvantages of the neural network approach already here. One topic is certainly the difficulties caused by changes in the sensor characteristics and Level 1 algorithms. Refer to experiences from other operational NRT services.

In the new version of the introduction we have cited the ASCAT NRT processing chain and that it is actually used for data assimilation at ECMWF. At the end of the same paragraph, we also comment on the fact that ASCAT and SMOS NRT processing chains are based on models whose parameters are fixed offline using a large amount of data and that those parameters should be updated if there are significant changes in the input data.

Page 4, line 5: What exactly do you mean by "arboreous component"? Do you mean the forest canopy?

We meant "trees". We replaced "arboreous" by "trees".

Page 4, line 16: ". . . was obtained by training . . ."

Done

Chapters 3.1 and 3.2: Please explain why the neural network relies on normalised data instead of the absolute brightness temperature values.

The local normalized index (or linear expectation) were showed to improve the retrieval results by Rodriguez-Fernandez et al. (2015). As explained in that paper, the use of those indexes as predictors was inspired by the "change detection" approach used by scatterometers. In the new version of the manuscript those informations are reminded explicitly in Section 3.1.

Page 8, line 23: 50 % is a very large value. Please explain.
Page 8, line 32: Here you allow no open water (0 %), which is in stark contrast to the 50 % threshold from above.

Thank you for pointing this out. Actually no additional filters regarding the water fraction were used to train the NN. The sentence has been removed. The maximum of 50 % of open water was used because it is the same value for ECMWF land products. Even with less than 50 % of open water the SM retrievals can be overestimated. However, L2SM retrievals are needed to define the local linear expectation index I, and the total number of L2SM retrievals with a free water fraction higher than 20 % is very low. Therefore, the same applies to the NRT NN retrieval. Still, SM values provided for footprints with open water are less reliable but it was decided to keep the values to allow the users to evaluate and decide by themselves. In the new version of the manuscript the readers are directed to the sea land surface mask aggregated into the ISEA grid in order to filter out mixed footprints if needed.

Page 8, bottom: Reformat list of put into table.

We guess that the referee remark was due to the double spacing of the "review" format for the manuscript but we have reformatted the list into a Table.

Chapter 4.1: Please explain why you decided not to use more advanced metrics.

We are not sure what are the metrics that the referee was thinking of. However, this paper concerns the presentation of the processor and the first evaluation results. In this context, we reckon that a global evaluation comparing the means of the two data sets, the bias, the Pearson correlation, RMSD and STDD, gives a good first idea of the properties of the new dataset. In addition, the evaluation of those two datasets has been compared to in situ measurements using the former metrics plus, in addition, the correlation of the anomalies time series to get further insight into the abilities of each dataset to capture the short term dynamics. We do reckon that the manuscript contains a good overview of quality metrics.

Discussion of Table 1 and Figure 5: The fact that the correlation R is overall slightly better for NRT than the L2 processor is noteworthy. Please discuss this in more detail and provide possible hypothesis why this is the case.

The reviewer is right. The NRT product shows a lower STDD and a higher R for the central two quartiles of the distribution (green boxes in Fig. 5). This behaviour was already found in previous studies such as Rodriguez-Fernandez et al. 2015. In the new version of the manuscript this result is discussed and interpreted in Sect. 5.3. Provided that the training is done with a large number of statistically representative samples, the NN will not be significantly affected by outliers or inconsistent values during the training phase and the NN output will the most likely (in the sense of the Bayes theorem) SM value taking into account a given set of input data. Thus, a good NN model can show slightly better quality metrics when compared to in situ measurements than the dataset

used to as reference to train the NN.

Page 16, line 15: Explain what you mean by "similar". What are the differences in implementation to the approach introduced by Rodriguez-Fernandez (2015)?

The main differences are :
- the training data is SMOS L2 and not ECMWF models
- a limited number of incidence angles is used in order to increase the swath width.
- Input Tbs are NRT Tbs and not Level 3 Tbs

Taking into account a comment by reviewer 1, which apparently understood that the NN was trained on ECMWF as in Rodriguez-Fernandez et al. 2015, we have decided to better explain the differences in Sect. 3 and to remove the sentence in the summary that could lead to misunderstandings for people not reading the whole paper.

---

## Editor Comment (EC1) · G.H. de Rooij (Editor) · 28 Jul 2017

Dear authors,

Three reviewers all gave constructive and essentially positive reviews with suggestions and recommendations that seem to be manageable. From the replies to the reviews it is clear that you too do not see any major road blocks.

I therefore kindly request you to revise the paper according to the suggestions offered by the reviewers, clarify those parts of the text that created some confusion, and clean up the typos.

[Figure]

I will probably ask at least one of the reviewers to go over your revision.

Yours sincerely,

Gerrit de Rooij

—————————————————

---

## Author Comment (AC2) · 28 Jul 2017

We thank the referee for his/her interesting and constructing comments.

Regarding comment 1.

Unfortunately it seems that there is a misunderstanding. ECMWF SM is not used to train the NN. The NN is trained on SMOS L2 SM. We will clarify this in a new version of the manuscript explaining better the differences of the approach used to implement the NRT SM product with respect to Rodriguez-Fernandez et al. (2015). Regarding the more general comment on the sensing depth, it is well-known problem to validate

remote sensing data using ground measurements. Unfortunately, this effect cannot easily be disentangle of the spatial representativity effect, that is, that remote sensing measurements are representative of 40-50 km while the in situ measurements are point-like measurements. This spatial representativeness can easily be a more significant effect than the sensing depth issue. In a new version of the manuscript we will add a discussion on both the sensing depth and the spatial representativeness issue. That said, the goal of this paper, as stated in the title, is to present the algorithmics and "first validation results". Follow up more thoughtful validation studies will be recommended, ideally from independent teams of potential users of this new ESA operational product.

Regarding comment 2

First, as said above, the SMOS NRT SM does not use ECWMF 0-7 cm soil moisture. Second, the goal of this study is not to modify the way the SMOS L2 SM algorithm deals with soil temperature, but to find a statistical alternative providing results faster at at least of the same accuracy, and we showed that the accuracy of the NRT is slightly increased with respect in situ measurements. In this context, we are afraid that a discussion of the effect of soil temperature in the Tor Vergata model (Dente et al.), would be out of the scope of this manuscript. As the referee reminds, the SMOS L2 SM uses the approach of Wigneron et al. to compute a soil effective temperature, and the temperature data used in the SMOS L2 SM algorithm comes from ECMWF model simulations. Therefore, it is logical to use ECMWF 0-7 cm temperature as a predictor in the input of a neural network trained on SMOS L2 SM. The evaluations after training, show that adding this soil temperature to complement the SMOS brightness temperatures improve the NN performances to capture the dynamics in the training data by $\sim$ 3 %. In a corrected version of the manuscript we will clarify this point taking into account the referee comment.

Typos

Thank you for pointing out those typos and misspellings. They will be corrected.

---

## Author Comment (AC3) · 28 Jul 2017

We thank the referee for his/her constructive comments.

Regarding the training and validation periods, the referee is completely right that the best approach is to perform the evaluation in a different period with respect to the training. That's exactly what we did. The period cited by the referee (June 2010-June 2012) is the period used for the training. As stated in section 4 of the manuscript the evaluation was done in 2015. Taking into account the referee comment, we reckon that we should stress this a bit more in a corrected version of the manuscript, maybe in section 2 or even in the abstract.

[Figure]

Thanks for the English editing and typos corrections.

---

## Author Response (AR1)

**Referee 1**

The manuscript reports a new effort in developing an SMOS NRT SM product by using a neural network approach. The NN algorithm uses six SMOS brightness temperatures at incidence angles from 30 to 45 deg binned at 5 deg intervals for horizontal and vertical polarizations and ECMWF IFS soil temperature at 0-7 cm depth. Snow depth and soil temperature at 0-7 cm depth < 274K from ECMWF IFS are also used to exclude certain (snow covered and frozen) situations.
The NRT SM product was compared to ESA SMOS L2 SM product and to the in-situ data from the SCAN and the USCRN soil moisture networks in USA and satisfactory statistics were reported. The used ECMWF soil temperature is from the layer of 0-7 cm, and the in-situ soil moisture sensors are installed at 5 cm depth.

Major comments
The used methods, data and assumptions are described in sufficient details with results of comparison reported and conclusions drawn. However in discussion of the results, some more in depth analysis of the differences and uncertainties between the different products would be very useful for the use of the current product.

We thank the referee for his/her interesting and constructive comments.

1. The depth of the retrieved SM SMOS NRT SM used ECMWF SM at 0-7 cm to train the NN, thus its retrieved SM should represent SM at the same depth, while ESA SMOS L2 SM represents that of the emission depth or sensing depth. In discussing the differences in between both products and those to the in-situ SM which is measured by sensors installed at 5 cm depth and represents an averaged SM around 5 cm, it would be important to point out such issues. In particular it would be important to explain why the correspondence is less good where very low correlation coefficients are reported and where and when such cases occur.

Unfortunately there is a misunderstanding. ECMWF SM is not used to train the NN. The NN is trained on SMOS L2 SM. In the new version of the manuscript we have explained better the differences of the approach used to implement the NRT SM product with respect to Rodriguez-Fernandez et al. (2015). This has been done in the first paragraph of Sect. 3.

Regarding the more general comment on the sensing depth, it is a well-known problem to validate remote sensing data using ground measurements. In addition, this effect cannot easily be disentangle of the spatial representativeness effect, that is, that remote sensing measurements are representative of 40-50 km while the in situ measurements are point-like measurements. This spatial representativeness can easily be a more significant effect than the sensing depth issue. In the new version of the manuscript we added a discussion on both the sensing depth and the spatial representativeness issue at the beginning of Sect. 4.2. In addition, we remind that the goal of this paper, as stated in the title, is to present the algorithmic and "first validation results" for the SMOS NRT SM product comparing to the Level 2 SMOS SM product. Therefore, spatial representativeness or sensing depth issues will not affect the comparison. In any case, follow up and more thoughtful validation studies of the SMOS NRT SM product are recommended, ideally from independent teams of potential users of this new ESA operational product.

2. The effective soil temperature
SMOS NRT SM uses ECMWF 0-7 cm soil moisture as its effective soil temperature. Previous

studies have concluded that the model temperature at this depth is not the most adequate one to use (e.g. Dente et al., 2014) and this would increase the uncertainties in particularly semi-arid and arid areas (e.g. Lv et al., 2016). It is also noticed that the SMOS L2 SM uses the Wigneron soil effective temperature. The authors need provide an analysis to settle this issue.

References:

Dente, L., et al. "Combined use of active and passive microwave satellite data to constrain a discrete scattering model." Remote Sensing of Environment 155 (2014): 222-238.

Lv, Shaoning, et al. "A reappraisal of global soil effective temperature schemes." Remote Sensing of Environment 183 (2016): 144-153.

First, as said above, the SMOS NRT SM does not use ECWMF 0-7 cm soil moisture. Second, the goal of this study is not to modify the way the SMOS L2 SM algorithm deals with soil temperature, but to find a statistical alternative providing results faster at at least of the same accuracy, and we showed that the accuracy of the NRT is slightly increased with respect in situ measurements. In this context, we are afraid that a discussion of the effect of soil temperature in the Tor Vergata model (Dente et al.), would be out of the scope of this manuscript. However, following the referee comment, in the new version we remind in Sect. 2.1.1 that he SMOS L2 SM uses the approach of Choudhury et al. 1982 with the parametrization by Wigneron et al. to compute a soil effective temperature, and that the temperature data used in the SMOS L2 SM algorithm comes from ECMWF model simulations. Therefore, it is logical to use ECMWF 0-7 cm temperature as a predictor in the input of a neural network trained on SMOS L2 SM. The evaluations after training, show that adding this soil temperature to complement the SMOS brightness temperatures improve the NN performances to capture the dynamics in the training data by ~ 3 %. In the corrected version of the manuscript we have also clarified this point taking into account the referee comment and adding those details in the first paragraph of Sect. 3.

Technical comments

While the manuscript is written in clear language, many typos need to be corrected.

Some are listed as follows:

P6L11: constrains -> constraints

P9L12: Levemberg-Marquard -> Levenberg-Marquardt

P9L18: this results -> these results

P11L5: please explain what is 'short-scale dynamics'

P12L16: for of -> for

P12L5: patters -> patterns

P12L6: spacial -> spatial

P14L24: shows –> show

P19L14: taken into account trough -> taken into account through

Thank you for this corrections. They have been taken into account.

**Referee 2**

This article details the performance of a neural network approach to soil moisture retrieval from SMOS brightness temperature measurements and ECMWF temperature estimates. Comparisons are made to the operational level 2 product and field measurements in northern America. A similar performance to the operational product is demonstrated, with a considerably lower lead time. The training and validation data appear to have been drawn from the same time period - June 2010-2012. This prompts questions as to the applicability of the approach to data acquired at different

times – if the system is only trained and validated for a certain temporal range, how does it perform on data outside that range? Five years of observations have been acquired since then, validation using some of these would address whether changes on the Earth's surface such as vegetation growth have an impact on accuracy, and whether the neural network approach is reliable when given inputs outside its trained range.

We fully agree with the reviewer. The evaluation period should be different to the training period. That's is exactly what we did. The NN was trained using data from June 2010 to June 2012. A pre-operational version of the NN was evaluated from May 2015 to November 2015 (Sect. 4). Taking into account the good results. The NN NRT SM product become operational in January 2016. In the corrected version of the manuscript the different periods are reminded in sect 2.1.2 .

Grammatical and spelling issues are detailed below. Figure 6(a) has either an outlier at the centre-right of the figure which needs to be explained, or a cursor which should be removed.

Thanks. Actually it was a cursor. The figure has been corrected.

Spelling - accesible should be accessible, "equipement" should be "equipment" Usage "arboreous" is not the right word; possibly "arboreal" was intended. p.8, l.32 repetition of "water" p. 9, l. 18 "this results" should be "these results" p.12 l.7 "well defined" should be "well-defined" p.16 l.14 remove additional "the"

Thank you. All those corrections have been done.

**Referee 3**

GENERAL COMMENTS
This is a well written and interesting paper describing a neural network approach for retrieving soil moisture from SMOS in near-real-time. The results are highly relevant for operational applications in hydrology, meteorology and other earth sciences. The results are realistic and I recommend publication after minor revisions.

We thank  the referee, Prof. Wagner, for his constructive comments that have allowed us to improve significantly the manuscript, in particular putting the new SMOS NRT SM in the broader context of other NRT SM processing chains such as the ASCAT one. We answer below the specific comments.

SPECIFIC COMMENTS
Page 3, lines 8-9: Explain why the NRT requirements cannot be met by the operational SMOS Level 2 processor. Is it just a matter of timeliness?

Actually the limiting factor is not the L2 processing chain but the total L0-L1C-L2 processing chain. The typical processing times to produce L1C brightness temperatures for a half-orbit is 1 hour. The L2 SM inversion can take up to 80 minutes if most of the grid points are land. However, some computations of the processing chain are synchronized with the ocean salinity chain and data handling and dissemination introduce overheads as well as the dissemination strategy was not designed for NRT delivery. Therefore, the typical latency time from acquisition to SM dissemination is around 6 hours.  This has been explained in the new version (page 3, 2$^{nd}$ paragraph). In contrast, since NRT Tbs are received by ECMWF it was decided to use them with a neural network algorithm for the NRT SM processing chain. A simplified version of the L2SM algorithm could also have been used with the NRT Tb's meeting almost the NRT requirement, but the NN is faster and much more simple to implement once the NN is trained.

Page 3: Please also discuss possible disadvantages of the neural network approach already here. One topic is certainly the difficulties caused by changes in the sensor characteristics and Level 1 algorithms. Refer to experiences from other operational NRT services.

In the new version of the introduction we have cited the ASCAT NRT processing chain and that it is actually used for data assimilation at ECMWF. At the end of the same paragraph, we also comment on the fact that ASCAT and SMOS NRT processing chains are based on models whose parameters are fixed offline using a large amount of data and that those parameters should be updated if there are significant changes in the input data.

Page 4, line 5: What exactly do you mean by "arboreous component"? Do you mean the forest canopy?

We meant "trees". We replaced "arboreous" by "trees".

Page 4, line 16: ". . . was obtained by training . . ."

Done

Chapters 3.1 and 3.2: Please explain why the neural network relies on normalised data instead of the absolute brightness temperature values.

The local normalized index (or linear expectation) were showed to improve the retrieval results by Rodriguez-Fernandez et al. (2015). As explained in that paper, the use of those indexes as predictors was inspired by the "change detection" approach used by scatterometers. In the new version of the manuscript those informations are reminded explicitly in Section 3.1.

Page 8, line 23: 50 % is a very large value. Please explain.
Page 8, line 32: Here you allow no open water (0 %), which is in stark contrast to the 50 % threshold from above.

Thank you for pointing this out. Actually no additional filters regarding the water fraction were used to train the NN. The sentence has been removed. The maximum of 50 % of open water was used because it is the same value for ECMWF land products. Even with less than 50 % of open water the SM retrievals can be overestimated. However, L2SM retrievals are needed to define the local linear expectation index I, and the total number of L2SM retrievals with a free water fraction higher than 20 % is very low. Therefore, the same applies to the NRT NN retrieval. Still, SM values provided for footprints with open water are less reliable but it was decided to keep the values to allow the users to evaluate and decide by themselves. In the new version of the manuscript the readers are directed to the sea land surface mask aggregated into the ISEA grid in order to filter out mixed footprints if needed.

Page 8, bottom: Reformat list of put into table.

We guess that the referee remark was due to the double spacing of the "review" format for the manuscript but we have reformatted the list into a Table.

Chapter 4.1: Please explain why you decided not to use more advanced metrics.

We are not sure what are the metrics that the referee was thinking of. However, this paper concerns the presentation of the processor and the first evaluation results. In this context, we reckon that a global evaluation comparing the means of the two data sets, the bias, the Pearson correlation, RMSD and STDD, gives a good first idea of the properties of the new dataset. In addition, the evaluation of those two datasets has been compared to in situ measurements using the former metrics plus, in addition, the correlation of the anomalies time series to get further insight into the abilities of each dataset to capture the short term dynamics. We do reckon that the manuscript contains a good overview of quality metrics.

Discussion of Table 1 and Figure 5: The fact that the correlation R is overall slightly better for NRT than the L2 processor is noteworthy. Please discuss this in more detail and provide possible hypothesis why this is the case.

The reviewer is right. The NRT product shows a lower STDD and a higher R for the central two quartiles of the distribution (green boxes in Fig. 5). This behaviour was already found in previous studies such as Rodriguez-Fernandez et al. 2015. In the new version of the manuscript this result is discussed and interpreted in Sect. 5.3. Provided that the training is done with a large number of statistically representative samples, the NN will not be significantly affected by outliers or inconsistent values during the training phase and the NN output will the most likely (in the sense of the Bayes theorem) SM value taking into account a given set of input data. Thus, a good NN model can show slightly better quality metrics when compared to in situ measurements than the dataset used to as reference to train the NN.

Page 16, line 15: Explain what you mean by "similar". What are the differences in implementation to the approach introduced by Rodriguez-Fernandez (2015)?

The main differences are :
- the training data is SMOS L2 and not ECMWF models
- a limited number of incidence angles is used in order to increase the swath width.
- Input Tbs are NRT Tbs and not Level 3 Tbs

Taking into account a comment by reviewer 1, which apparently understood that the NN was trained on ECMWF as in Rodriguez-Fernandez et al. 2015, we have decided to better explain the differences in Sect. 3 and to remove the sentence in the summary that could lead to misunderstandings for people not reading the whole paper.

[revised manuscript text omitted]